# Cytoophidia Maintain the Integrity of *Drosophila* Follicle Epithelium

**DOI:** 10.3390/ijms232315282

**Published:** 2022-12-04

**Authors:** Qiao-Qi Wang, Dong-Dong You, Ji-Long Liu

**Affiliations:** 1School of Life Science and Technology, Shanghai Technology University, Shanghai 201210, China; 2Department of Physiology, Anatomy and Genetics, University of Oxford, Oxford OX1 3PT, UK

**Keywords:** CTP synthase, cytoophidium, *Drosophila*, epithelium, follicle cell, ingression

## Abstract

CTP synthase (CTPS) forms a filamentous structure termed the cytoophidium in all three domains of life. The female reproductive system of *Drosophila* is an excellent model for studying the physiological function of cytoophidia. Here, we use *CTPS^H355A^*, a point mutation that destroys the cytoophidium-forming ability of CTPS, to explore the in vivo function of cytoophidia. In *CTPS^H355A^* egg chambers, we observe the ingression and increased heterogeneity of follicle cells. In addition, we find that the cytoophidium-forming ability of CTPS, rather than the protein level, is the cause of the defects observed in *CTPS^H355A^* mutants. To sum up, our data indicate that cytoophidia play an important role in maintaining the integrity of follicle epithelium.

## 1. Introduction

CTP synthase (CTPS) is a glutamate aminotransferase that catalyzes the transfer of amide nitrogen from glutamine to the C-4 position of UTP. CTP, the product of CTPS, is an important nucleotide and is a component of the synthesis of RNA, DNA, and sialoglycoprotein. It also acts as an energy coupler for some metabolic reactions, such as the synthesis of glycerophospholipids and glycosylated proteins [1,2].

In 2010, CTPS was found to form filamentous structures termed cytoophidia in *Drosophila* [3]. Subsequently, CTPS has been found to form filamentous structures in bacteria [4] and *S. cerevisiae* [5]. In the following years, the existence of cytoophidia was confirmed in human cells [6], *S. pombe* [7], *Arabidopsis thaliana* [8], and archaea [9], which indicates that cytoophidia are highly conserved in evolution.

Compartmentation is the basis for the function of organelles [10]. The classical cellular compartmentation in eukaryotic cells is achieved through membrane-bound organelles, such as the endoplasmic reticulum, mitochondrion, and Golgi apparatus [11]. Compartmentation establishes a physical boundary for the biological processes within cells, enabling cells to carry out different metabolic activities at the same time, generate specific microenvironments, regulate biological processes in time and space, and determine the specific location where biological processes should occur. The formation of cytoophidia realizes the regionalization of CTPS, and its location in cells may therefore have corresponding physiological significance.

Cells in the *Drosophila* ovary exhibit vigorous anabolic activity because they need nutrients for development. Cytoophidia are observed from region 2 of the germarium to stage 10A of oogenesis. Based on the widespread presence of cytoophidia in germline cells and follicle epithelial cells of *Drosophila* ovaries [12], and the characteristics of cytoophidium observed in germline cells and follicle epithelial cells at most stages of oogenesis, *Drosophila* ovarian follicle cells have become a classic model for studying cytoophidia.

Epidermal tissues form the boundaries of organs, where they perform a range of functions, including secretion, absorption, and protection. These tissues are usually composed of discrete cells, forming a single-cell thick sheet. Follicle epithelium is a simple epithelium. In the process of division, the cells of simple epithelium have a specific orientation of the spindle, so that both daughter cells are located in the epithelial plane. This is considered to be very important for maintaining the integrity of follicle epithelium and preventing hyperplasia [13,14,15,16].

An egg chamber consists of hundreds of follicle cells, and each follicle cell has multiple membrane domains including apical, basal, and lateral. The end adjacent to germline cells is defined as the apical side, and the end far away from germline cells is defined as the basal side [17]. Before mitosis, follicle cells will move toward the apical direction, which may be caused by the extrusion of neighbor cells. This movement results in the displacement of some cells from the tissue layer. Usually, the displaced cells need to be reintegrated to support tissue growth and maintain tissue architecture [18,19,20].

In a previous study, we find that cytoophidia are specifically distributed on the basolateral side of follicle cells, and this specific distribution is related to the polarity regulator of the cell membrane [21]. Therefore, we need to understand the function of cytoophidium which is specifically distributed in follicle cells.

In this study, we describe the effects of cytoophidium disassembly on follicle epithelium integrity. We are also concerned about whether these effects are directly related to the assembly of CTPS into cytoophidia, rather than to the level of CTPS protein. Our results indicate that cytoophidia play an important role in maintaining the integrity of follicle epithelium. In addition, we eliminate the influence of tissue-tissue interaction and find that cytoophidia can directly affect the integrity of follicle epithelium.

## 2. Results

### 2.1. CTPS Forms Cytoophidia in Drosophila Follicle Cells

Cells in *Drosophila* ovaries exhibit vigorous anabolic activity because they need nutrients for development. During *Drosophila* oogenesis, the follicle epithelium is a sheet of monolayer cells that encase germline cells. CTPS, as the synthase of CTP, plays an important role in the regulation of tissue growth and development. Cytoophidia exist in several different types of cells in the *Drosophila* ovary from region 2 of the germarium to stage 10A of oogenesis, including epithelial follicle cells (Figure 1A–C) and germline cells (Figure 1D–F).

### 2.2. Follicle Cells Undergo Ingression in CTPS^H355A^ Egg Chambers

The amino acid histidine at the 355th position, or His355, lies at the tetramer-tetramer interface of CTPS [22]. If the H355 site is mutated, the cytoophidium cannot be formed. Previous studies showed that the H355 site is essential for its polymerization, but not enzymatic function [23,24]. Our laboratory has solved the structure of *Drosophila melanogaster* CTPS (dmCTPS) and found that the H355 site lies at the tetramer–tetramer interface and does not affect the catalytic site [25]. Therefore, we constructed an H355A point-mutated knock-in *Drosophila* strain to investigate whether the disassembly of cytoophidia would affect follicle cells. Former studies found that the H355A served as a dominant negative point mutation [26] (Appendix A). 

In order to find out whether the abnormality is caused by the inability of CTPS to aggregate due to H355A point mutation or the addition of mCherry tag, our laboratory constructed another *Drosophila* strain with mCherry added to the C-terminus of CTPS based on *w^1118^*. To determine whether the feature of cytoophidium localization was in fact introduced by protein fusion between CTPS and mCherry tag, we performed immunofluorescence microscopy and directly detected the CTPS protein of the *w^1118^* fly and found no difference [27]. It is proven by the observation that the knock-in mCherry tag does not affect the polymerization of CTPS protein. The morphology of the *CTPS-mCherry^KI^ Drosophila(CTPS-mCh)* ovaries is consistent with that of the *w^1118^*, which implies that the *CTPS-mCherry^KI^ Drosophila* can also be used as control in our experiment (Figure 2A–F). Besides, our laboratory has used the *CTPS-mCherry^KI^ Drosophila* as control in previous studies [28,29].

When constructing the point-mutated *Drosophila* strain, we added a mCherry tag at the C-terminal of CTPS. Through confocal microscopy, we observed diffuse mCherry signal in *Drosophila* follicle cells, which confirmed that the CTPS could not form the cytoophidium after mutation at the H355 site (Figure 3A,F). In the egg chamber of wild-type flies, follicle cells are monolayer epidermal cells. We observed their morphological characteristics by immunofluorescence staining. The cell membrane was labeled with an antibody against Armadillo. We found that in the egg chamber of *CTPS^H355A/H355A^-mCherry* knock-in homozygous fly (hereinafter referred to as *CTPS^H355A^* strain), some follicle cells originally arranged in a monolayer migrated inward (ie. ingression), thus disrupting the monolayer arrangement. The ingression of follicle cells occurs not only in the early stages of oogenesis, such as stage 5 (Figure 3A–E), but also in the middle stages of oogenesis, such as stage 8 (Figure 3F–J).

Our study mainly focused on stage 8 egg chambers. We demonstrated the ingression of follicle cells in stage 8 egg chambers through three-dimensional reconstruction (Figure 4A–C). Combined with the morphological changes of follicle cells observed on the surface of the egg chambers, we speculated that the integrity of follicle epithelia would be disturbed when CTPS could not assemble into cytoophidia. Through statistical analysis, we found that approximately 20% of egg chambers at stage 8 have follicle cells ingression by counting 20 stage 8 egg chambers of each genotype (Figure 4D). Our results indicate that the widely and specifically distributed cytoophidia play a role in maintaining the integrity of follicle epithelia.

### 2.3. Ingressive Follicle Cells Display Abnormal DCAD2 Pattern

In a previous study, we found that cytoophidia are specifically located on the lateral and basal sides of follicle cells [21]. The polarity regulators of follicle cells and adherens junctions have certain effects on the maintenance of cytoophidia. To explore whether the cell membrane components of ingressive follicle cells would be affected when cytoophidium fails to form, we labeled the basolateral regulator Dlg of follicle cells, adherens junctions DE-Cadherin DCAD2, and cell membrane protein Hts. After immunostaining, in the follicle epithelium labeled with Hts and Dlg, there was no significant difference between the cell membrane of ingressive follicle cells and that of normal follicle cells (Figure 5A–H).

Under the condition of DCAD2 labeling, we found that DCAD2 showed an abnormal pattern in the ingressive follicle cells compared with normally arranged follicle cells. In normal follicle cells, the end near the germline cells is defined as the apical side and the end near the muscle layer is defined as the basal side. From the cross-sectional view of the lateral side of the stage 8 egg chamber, the DCAD2 pattern should be adjacent to the germline cells. However, in the ingressive follicle cell, DCAD2 could be seen flipping in the direction rather than at the apical side (Figure 6).

### 2.4. CTPS^H355A^ Follicle Cells Increase the Heterogeneity

The follicle epithelium of *Drosophila* consists of a monolayer of follicle cells, which surround the oocyte and 15 nurse cells. Follicle cells gradually differentiate into various subpopulations, which will undergo morphological changes. After stage 6, the follicle cells cease mitosis and are arranged in a hexagonal pattern, which means that under normal circumstances, each follicle cell contacts six adjacent cells, most of which are hexagonal and well arranged on the surface of egg chambers. However, in *CTPS^H355A^* mutant, the assembly of cytoophidia was disrupted and the number of non-hexagonal cells increased. We segmented the cell by the membrane staining of Hts protein, counting the different shapes cell by cell. In *CTPS^H355A^* mutant, we observed many pentagonal follicle cells, and the heptagonal cells increased by about 10% (Figure 7A–D).

The quantification further confirmed that compared with *w^1118^*, there was difference in the number of heptagonal follicle cells in stage 8 egg chambers of the *CTPS^H355A^* mutant. Moreover, the number of hexagonal follicle cells in *CTPS^H355A^* egg chambers decreased by about 15% at stage 8, while the number of polygonal cells increased by about 10% (Figure 7E–G). Considering that the more sides the polygon, the closer it is to the round circle, we speculate that when cytoophidia cannot be formed, the cell membrane will be affected, and the tight arrangement of epithelial follicle cells will not be maintained. Morphology changes indicate that cytoophidia located at the basolateral side may play a role in maintaining the integrity of follicle epithelium.

### 2.5. Follicle Epithelia Reduce Compactness in CTPS^H355A^ Mutant

In the wild-type flies, follicle epithelial cells at stages 4–9 of oogenesis are tightly packed, and most of the hexagonal follicle cells enclose germ cells. In the case that CTPS could not be assembled into cytoophidia, we observed that closely arranged epithelial follicle cells became relatively loose, and follicle cells of similar size in the wild-type became relatively very large or very small, which was not conducive to compact arrangement (Figure 8A–L).

In order to further clarify the observed phenomenon, we segmented the follicle cell surface based on the cell membrane and calculated the basal area of each follicle cell on an egg chamber through software. The average area ratio of the group with the largest area of three adjacent follicle cells and the group with the smallest area of three adjacent follicle cells was used as an indicator of follicle cell heterogeneity. The higher the ratio, the higher the heterogeneity of surface follicle cell. The quantitative analysis showed that the average area of follicle cells at stage 8 *CTPS^H355A^* was smaller than that of the wild-type, but the heterogeneity was much higher than that of the wild-type (Figure 8M,N).

### 2.6. Follicle Cell Ingression Occurs in Egg Chamber Overexpessing CTPS^H355A^

In previous studies, our laboratory found that the formation of cytoophidia can prolong the half-life of CTPS protein in mammalian cells. Therefore, we want to know whether H355A point mutation affects CTPS protein level in *Drosophila* ovaries. Western blot results confirmed that the level of CTPS protein in *Drosophila* ovaries after *CTPS^H355A^* mutation was lower than that in the *wild-type* (Figure 9H,I). Thus, we want to investigate whether the phenotypes observed in the *CTPS^H355A^* strain are caused by the decrease of CTPS protein level.

To eliminate the influence of protein level, we used the *Actin-Gal4* driver to overexpress *CTPS^H355A^* in *Drosophila* ovaries. We constructed *Actin-Gal4-driven Drosophila* strains overexpressing *CTPS^H355A^* (*Actin > UAS CTPS^H355A^-mCherry-OE*) or *wild-type CTPS* (*Actin > UAS CTPS-mCherry-OE*). Western blot confirmed that there was no significant difference in CTPS level between the *Actin > UAS CTPS-mCherry-OE* heterozygous strain and the *Actin > UAS CTPS^H355A^-mCherry-OE* homozygous strain (Figure 9J,K).

We found that the distribution of cytoophidia in the basolateral side of follicle cells could be clearly observed in *Actin > UAS CTPS-mCherry-OE* heterozygous egg chambers (Figure 9A,B). Almost every follicle cell had one or two cytoophidia, and follicle cells were arranged in a single layer. In *Actin > UAS CTPS^H355A^-mCherry-OE* homozygous flies, the diffused distribution of CTPS^H355A^ could be observed, and the ingressive follicle cells appeared as well (Figure 9C–G). These results indicate that loss of the cytoophidium-forming ability of CTPS, rather than its protein level, is the primary cause of follicle cell ingression in the *CTPS^H355A^* mutant.

### 2.7. Overexpession of CTPS^H355A^ Increases the Heterogeneity of Follicle Cells

Similarly, we wanted to examine whether the heterogeneity of follicle cells was affected by the level of CTPS protein. According to our study, there was a long and curly cytoophidium in each follicle cell on the surface of *Actin > UAS CTPS-mCherry-OE* heterozygous egg chamber. Compared with the wild-type egg chambers, where cytoophidia are mostly rod-shaped and distributed along the cell membrane, the elongated cytoophidia were still distributed along the cell membrane after the overexpression of CTPS (Figure 10A–C).

The diffused distribution of CTPS was confirmed on the surface of the egg chamber of *Actin > UAS CTPS^H355A^-mCherry-OE* homozygous fly. The changes of morphology and cell size showed that the heterogeneity of follicle cells was enhanced because there was no cytoophidium on the cell membrane. It seemed that these follicle cells could not even be arranged tightly (Figure 10D–F).

### 2.8. Follicle Cell-Specific Overexpression of CTPS^H355A^ Impairs the Integrity of Follicle Epithelium

After excluding the influence of CTPS protein level, we wanted to further eliminate the effect of the inter-tissue interaction caused by the ubiquitous expression of *Actin-Gal4*. To this end, we constructed a strain overexpressing CTPS^H355A^ using the same *UAS CTPS^H355A^-mCherry-OE* and *UAS CTPS-mCherry-OE* strains together with *Tj-Gal4* specifically expressed in follicle cells. As a control, *wild-type CTPS* was overexpressed specifically in follicle cells using the *Tj-Gal4* driver. Our results confirmed that the integrity of follicle epithelium was impaired when *CTPS^H355A^* was overexpressed specifically in follicle cells (Figure 11A–N).

### 2.9. Space between Muscle Sheath and Egg Chamber Increases in CTPS^H355A^

IF is a member of the integrin complex and widely exists in the muscle layer that encloses the ovarioles. We found that in the wild-type ovary, the muscle sheath tightly wrapped the ovarioles and drove their movement (Figure 12A,B), which was conducive to common life activities such as oogenesis. In *CTPS^H355A^* ovaries, the space between the muscle sheath and the egg chamber was significantly increased, and the egg chamber almost collapsed from the muscle sheath (Figure 12C–E), which might affect normal physiological activities.

## 3. Discussion

To explore the physiological function of cytoophidia in *Drosophila* follicle cells, we analyze the changes in follicle cells in CTPS mutant when cytoophidia cannot be formed. Our results indicate that the integrity of follicle epithelium is compromised when CTPS lose its cytoophidium-forming capability.

In this study, we generate transgenic flies with a point mutation in CTPS. Mutations do not affect enzymatic activity but lead to the disassembly of cytoophidia. In the mutant flies, the integrity of follicle epithelia is impaired with two related phenotypes: (1) ingression of follicle cells and (2) heterogeneous follicle cells.

We have previously discovered that cytoophidia are specifically distributed on the basolateral side of follicle cells [21]. Moreover, when the polarity of follicle cells is disrupted, cytoophidia will become unstable, especially due to the disruption of apical regulators. In this study, the apical polarity of follicle cells is indeed affected by the absence of cytoophidia. Our data indicate that the cytoophidium, as a kind of membraneless organelle, maintains its specific subcellular localization in biological processes.

In these experiments, we also notice that cytoophidia play a role in maintaining the integrity of follicle epithelium. We speculate that cytoophidia located at the basolateral side of follicle cells may play a role in supporting follicle cells. In the absence of cytoophidia, the mechanical tension of the follicle cell membrane will be reduced, making it more difficult to maintain the cell morphology. Therefore, follicle cells are more likely to be drawn into polygons and expanded by surrounding cells or squeezed and reduced by surrounding cells. Similarly, due to the weakening of membrane mechanical tension, the follicle cells migrating inward after mitosis cannot be reintegrated into the follicular monolayer, resulting in the ingression. Our laboratory also found that in the male reproductive system of *Drosophila*, when CTPS cannot form cytoophidia, the main cells on the surface of the accessory gland may be difficult to maintain their cell shape, and two horizontally arranged nuclei appear to be vertically arranged. This further support our hypothesis [30]. 

When cytoophidia are disassembled, the observed separation of egg chamber and muscle sheath may also be due to the disappearance of the supporting force of cytoophidia. When cytoophidia cannot be formed, the internal supporting force of each follicle cell is weakened, leading to the collapse of the entire egg chamber. Considering that the follicle epithelium will develop into the eggshell of a fertilized egg in the later stages [31], it is possible that its shell hardness and the hatchability of the fertilized egg will also be affected accordingly.

However, we could not simply rescue the phenotypes found in the *CTPS^H355A^-mCh* mutant by expressing CTPS-mCherry protein. Our previous studies in mammals [26] and *Drosophila* [27] confirmed that the CTPS^H355A^ point mutation is dominant-negative, which is to say that as long as the CTPS^H355A^ protein exists, the CTPS protein would not be able to assemble into cytoophidium [26,27]. Because the H355A point mutation of CTPS would disrupt the assembly of the cytoophidia dominant negatively, we have analyzed the *CTPS^H355A/TM6B^-mCh* egg chambers and found that *CTPS^H355A/TM6B^* also have defects in follicle epithelial integrity mentioned above (Appendix A). These results further validate our hypothesis that the cytoophidium structure plays a certain role in maintaining epithelial integrity, and the dominant negative CTPS point mutation confirmed that it is crucial for the assembly of cytoophidium.

Since the first discovery of cytoophidia in our laboratory in 2010, great progress has been made in the research on the existence of cytoophidia in different species and different types of cells. However, knowledge concerning the function of this new type of organelle widely existing in organisms is still in the initial stage. Therefore, our work has potential reference value for understanding the role of cytoophidia in *Drosophila* follicle cells. Our results indicate that forming cytoophidia is crucial to epithelial integrity.

## 4. Materials and Methods

### 4.1. Fly Stocks

All stocks were maintained at 25 °C on standard cornmeal. Both *w^1118^* and C-terminal mChe-4V5 tagged CTPS knock-in flies out of *w^1118^* produced in our laboratory were used as wild-type controls unless stated otherwise. The stocks used were: (1) *CTPS^H335A^ mutated with mChe-4V5 tagged CTPS Knock-in fly*, (2) *Actin-Gal4/Cyo* (A gift from Guanjun Gao’s lab, ubiquitous expression under strong promoter, a chromosome II insertion balanced over Curly of Oster [32]), (3) *Tj-Gal4* (A gift from Kun Dou’s lab [33,34]), (4) *Sp/Cyo; Sb/Tm6B* (Institute of Biochemistry and Cell Biology, Chinese Academy of Sciences, Drosophila Resources and Technology Platform), (5) *UAS CTPS-mCherry-OE, and UAS CTPS^H355A^-mCherry-OE* [27].

### 4.2. Fly Genetics

Transgenic flies expressing full-length CTPS (isoform C, LD25005) under the UAS promoter (*UAS-CTPS*) were generated in our lab (Back cross 5 generations before use) [35]. To generate the *Tj > UAS CTPS-OE*, we crossed the homozygous *UAS CTPS-OE Drosophila* and *Tj-Gal4 Drosophila* with the double balancer *Drosophila* for one generation, and the target generation was inbred for two generations to get the homozygous *Tj-Gal4*; *UAS CTPS-mCherry-OE Drosophila*. Same as the Tj-Gal4; *UAS CTPS^H355A^-mCherry-OE; Actin-Gal4; UAS CTPS-mCherry-OE* and *Actin-Gal4*; *UAS CTPS^H355A^-mCherry-OE* strains.

### 4.3. Generation of Transgenic Flies

For polymerase chain reaction (PCR), PUASTattb plasmids were used as the template, and phanta Maxa Super-Fidelity DNA Polymerase (Vazyme, #P505) as the polymerase. Sequences for primers were as below:

H355A-F: GAGCAAGTACGCCAAGGAGTGGCAGAAGCTATGCGATAGCCAT;

H355A-R: TGCCACTCCTTGGCGTACTTGCTCGGCTCAGAATGCAAAGTTT

After obtaining the required plasmids, the *CTPS^H355A^ Drosophila* strain was constructed by microinjection.

### 4.4. Immunohistochemistry

Ovaries from flies were dissected in Grace’s Insect Medium (Gibco) and then fixed in 4% formaldehyde (Sigma) diluted in PBS for 10 min before immunofluorescence staining. The samples were then washed twice using PST (0.5% horse serum + 0.3% Triton × 100 in PBS). For membrane staining, samples were incubated with primary antibodies at room temperature overnight, and then washed using PST. Secondary antibodies were used to incubate the samples at room temperature for another night.

Primary antibodies used in this study were rabbit anti-CTPS (1:1000; y-88, sc-134457, Santa Cruz BioTech Ltd., Santa Cruz, CA, USA), mouse anti-Discs Large (1:500, Developmental Studies Hybridoma Bank, Iowa City, IA, USA), mouse anti-D-E Cadherin (1:500, Developmental Studies Hybridoma Bank), mouse anti-HTS (1:1000, Developmental Studies Hybridoma Bank, Cat. No. AB_528070), mouse anti-Armadillo (1:500, Developmental Studies Hybridoma Bank). Secondary antibodies used in this study were anti-mouse, rabbit, or goat antibodies that were labeled with Alexa Fluor^®^ 488 (Molecular Probes), or with Cy5 (Jackson ImmunoResearch Laboratories, West Grove, PA, USA). Hoechst 33342 was used to label DNA.

### 4.5. Microscopy and Image Analysis

All images were obtained under laser-scanning confocal microscopy (Zeiss 880). Image processing was performed using Zeiss Zen. ImageJ was used to analyze the area and number of follicle cells.

We used the ImageJ SCF to segment the follicle cells by the membrane, then use ImageJ cell counter to calculate different shapes of cells to get the number of polygonal cells. We used ImageJ to measure the area of each cell. For each statistical quantification, we collected the surface images using Zeiss 880 with the interval as 0.5 μm for z-stack, 5 stage 8 egg chambers were quantified per genotype, biological repeats = 3. Mann-Whitney U test was conducted for comparison.

### 4.6. Western Blotting

Female adult ovaries of *Drosophila* were collected with gathered into lysis buffer RIPA (Meilunbio, Dalian, China) with protease inhibitor cocktail (Bimake, Shanghai, China) for Western blotting, and then ground with 1 mm Zirconia beads in Sonicator (Shanghai Jing Xin, Shanghai, China). The sample would then lysis on ice for up to 30 min. Samples were centrifuged for 10 min at 10,000 g at 4 °C. The 6× protein loading buffer was pipetted into the supernatants and boiled at 99 °C for 15 min to obtain protein. Then, the protein sample was run through 10% SDS-PAGE gels and transferred to PVDF membranes. At room temperature, membrane was incubated with 5% *w*/*v* nonfat dry milk dissolved by 1× TBST for 1 h of blocking. Then, the membrane was incubated with primary antibodies in 5% *w*/*v* nonfat milk at 4 °C and gently shaken overnight.

The following primary antibodies were used in this study: anti-mCherry Tag Monoclonal antibodies (Cat. No. A02080, Abbkine, Beijing China), mouse anti-a-Tubulin antibodies (Cat. No. T6199, Sigma). The membranes were washed three times for 5 min per time with shaking, then incubated with secondary antibodies (anti-mouse IgG, HRP-linked antibody, Cell Signaling, Danvers, MA, USA) diluted in 5% *w*/*v* nonfat milk at room temperature for 1 h. An Amersham Imager 600 (General Electric, Boston, MA, USA) and Pierce ECL Reagent Kit (Cat. No. 32106, Thermo Fisher, Waltham, MA, USA) were adopted for the chemiluminescence immunoassay. Protein levels were quantified on ImageJ (National Institutes of Health, Bethesda, MD, USA) and normalized to tubulin. At least three biological replicates were quantified.

### 4.7. Data Analysis

Images collected by confocal microscopy were processed using Adobe Illustrator and ImageJ. Cell segmentation based on the cell membrane was achieved using CellPose and SCF methods. Quantitative analysis was processed by Excel and GraphPad. The Mann–Whitney U test was conducted to get the *p*-value.

## Figures and Tables

**Figure 1 ijms-23-15282-f001:**
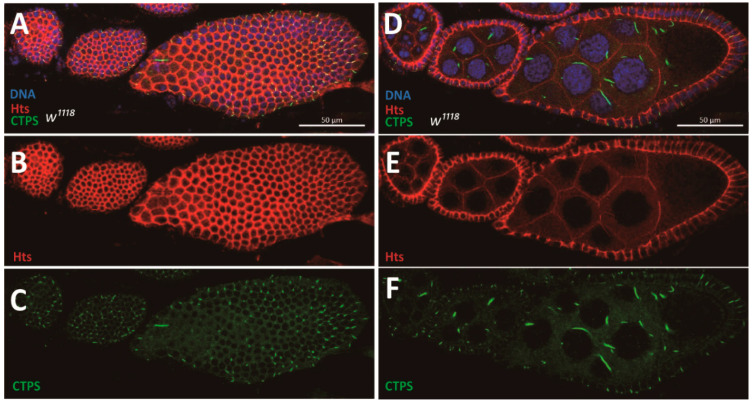
**Follicle cells maintain monolayer during *Drosophila* oogenesis.** (**A**–**C**) Surface view of a wild-type ovariole containing different stages of egg chambers. The ovariole is subjected to immunofluorescence analysis with antibodies against CTPS (green) and Hts (red, labelling cell membranes). DNA is labelled by Hoechst 33342 (blue). (**A**) Cytoophidia are distributed almost at different stages of each follicle cell. (**B**) The boundaies of follicle epithelia are displayed by a single projection of Hts staining. (**C**) CTPS staining shows the distribution of cytoophidia on the surface of egg chambers. (**D**–**F**) Side view of the same ovariole in (**A**–**C**). (**D**) Monolayer follicle cells envelop germline cells. (**E**) A single projection of Hts staining shows the monolayer structure of follicle epithelia. (**F**) CTPS staining shows the distribution of cytoophidia in follicle cells and germline cells. Scale bars, 50 μm.

**Figure 2 ijms-23-15282-f002:**
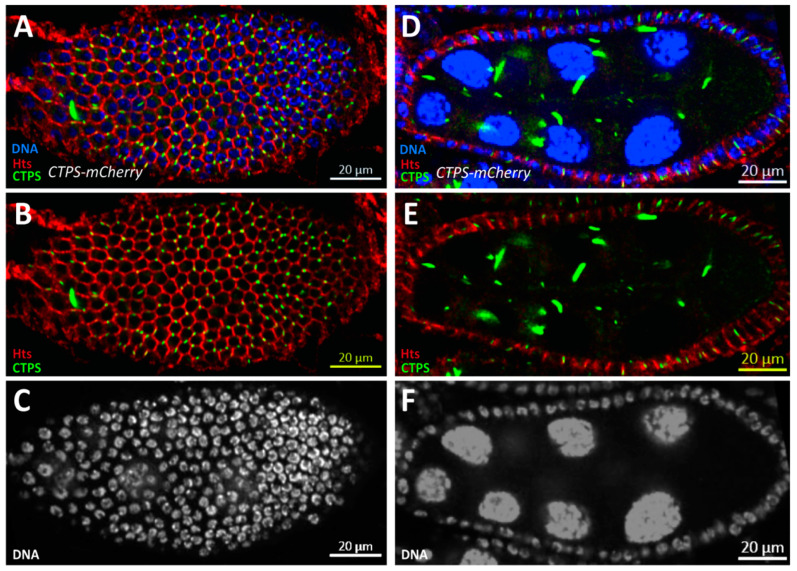
***CTPS-mCh* ovaries have same morphology as wild-type ovaries.** (**A**–**C**) Surface view of a stage 8 CTPS-mCh egg chamber. The CTPS signal (green) shown is obtained using mCherry-tagged CTPS. Hts (red) staining marks cell membranes and Hoechst (blue) for DNA. (**A**) mCherry moiety doesn’t affect CTPS assembly. (**B**) Merged panel of Hts and CTPS to display the cytoophidia location. (**C**) Single panel of the nucleus. (**D**) Lateral view of a stage 8 *CTPS-mCh* egg chamber. (**E**) Merged panel of Hts and CTPS to show the monolayer structure of follicle epithelia as well as cytoophidia distribution. (**F**) Single panel of nucleus to stress the single-layer follicle epithelia.

**Figure 3 ijms-23-15282-f003:**
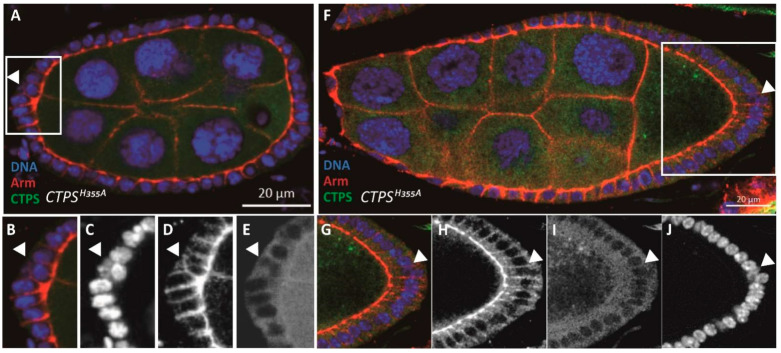
**Follicle cell ingression in *CTPS^H355A^* mutant in early and middle stages of oogenesis.** (**A**) Cross section of a *CTPS^H355A^* stage 6 egg chamber. The egg chamber is labeled with CTPS (green), Armadillo (red) for apical complex in follicle cells and Hoechst 33342 (blue) for DNA. The white rectangle emphasizes the ingression of a follicle cell. (**B**–**E**) Close-up images of the ingressive follicle cell, indicated by the yellow arrow. No cytoophidium is formed due to the H355A mutation in CTPS. (**F**) Cross section of a *CTPS^H355A^* egg chamber at stage 9. The white rectangle emphasizes the ingression of a follicle cell. (**G**–**J**) Close-up images of the ingressive follicle cell, indicated by the yellow arrow. Scale bars, 20 μm.

**Figure 4 ijms-23-15282-f004:**
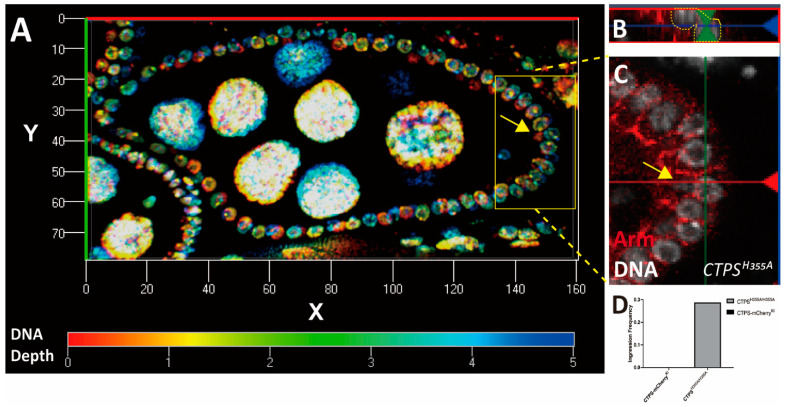
**Three dimensional view of follicle cell ingression.** (**A**) Cross section of a three-dimensional *CTPS^H355A^* egg chamber. A three-dimensional view of DNA stacked in layers. The interval between each layer is 0.5 μm, and a total of 12 layers are superimposed. The color from red to blue indicates the depth of DNA. (**B**) Side view of an ingressive nucleus. (**C**) On the xz plane, with the yellow dotted line marking the ingressive cell. (**D**) Quantification of the ingression frequency, 20 stage 8 egg chambers were counted per genotype.

**Figure 5 ijms-23-15282-f005:**
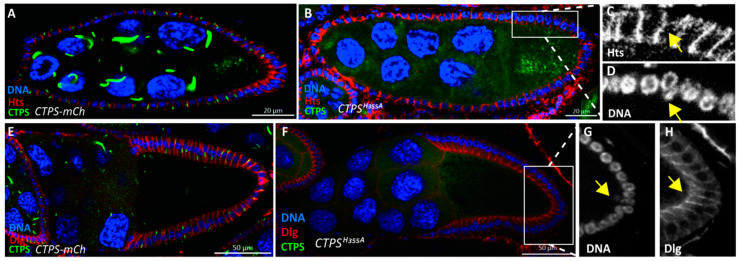
**Ingression of follicle cells labeled with different membrane proteins.** (**A**) Lateral view of a stage 8 egg chamber. Hts (red) labels the hulitaishao protein mainly presents at the lateral of follicle cell membranes. mCherry-tagged CTPS is shown green, and Hoechst (blue) for DNA. (**B**) A stage 8 egg chamber of *CTPS^H355A^ Drosophila*. mCherry-tagged *CTPS^H355A^* is diffused. (**C**) The Hts pattern of an ingressive follicle cell. (**D**) Yellow arrow pointed out the ingressive cell nuclear. (**E**) Lateral view of a stage 10A egg chamber. Dlg (red) labels the discs large protein, which presents in the lateral and basal side of follicle cell. CTPS-mCherry cytoophidia can be observed, and Hoechst (blue) for DNA. (**F**) Lateral view of the *CTPS^H355A^* egg chamber at stage 10A. (**G**) The ingressive cell nuclear is stressed by the yellow arrow. (**H**) The Dlg pattern of the follicle cell ingression.

**Figure 6 ijms-23-15282-f006:**
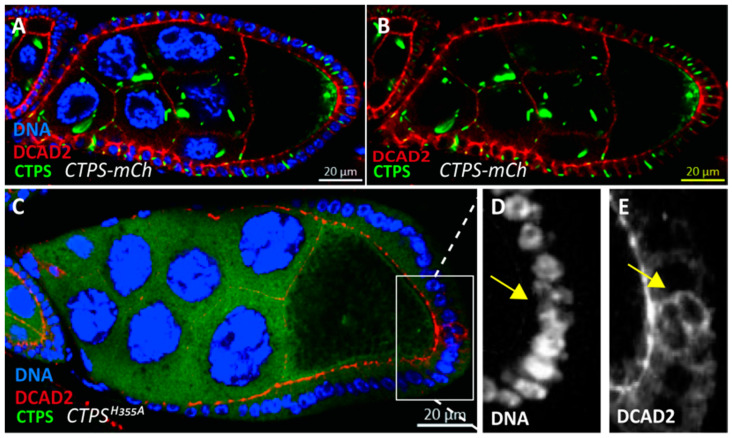
**DCAD2 distribution is disturbed in *CTPS^H355A^* follicle epithelia.** (**A**) Cross section of Drosophila egg chamber expressing CTPS labelled by mCherry. The part of the follicle cell adjacent to the nurse cell is called apical, and the DE-Cadherin labeled by DCAD2 (red) is located at the apical of follicle cells. (**B**) DCAD2 together with CTPS as control to show the normal distribution of DCAD2. (**C**) Lateral view of a *CTPS^H355A^–mCh* egg chamber with abnormal follicle cell. (**D**) Yellow arrow pointed to the follicle cell ingression. (**E**) Ingressive folllicle cell pointed to by yellow arrows show abnormal distribution of DCAD2.

**Figure 7 ijms-23-15282-f007:**
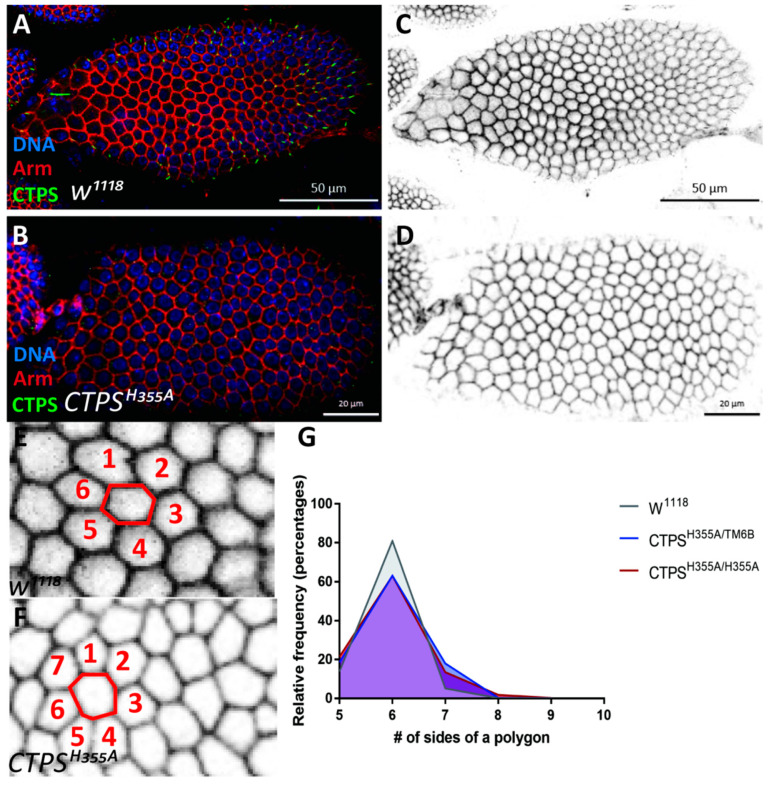
**Morphological comparision of follicles between wild-type and *CTPS^H355A^*.** (**A**–**F**) Surface view of stage 8 egg chambers. Membrane was labeled by Armadillo (red), and CTPS is shown in green. (**A**,**C**) *w^1118^* egg chamber. Scale bars, 50 μm. (**B**,**D**) *CTPS^H355A^* egg chamber. Scale bars, 20 μm. (**E**) Zoom-in view of (**C**). (**F**) Zoom-in view of (**D**). Redlines in E and F outline a central follicle cell with neighboring follicle cells in numbers. Note that the number of neighboring cells reflects the number of sides of the central polygonal follicle cells. (**G**) Quantitative analysis of the morphological difference between the wild-type control and *CTPS^H355A^* follicle cells (6 egg chambers were quantified per genotype, biological repeats = 3).

**Figure 8 ijms-23-15282-f008:**
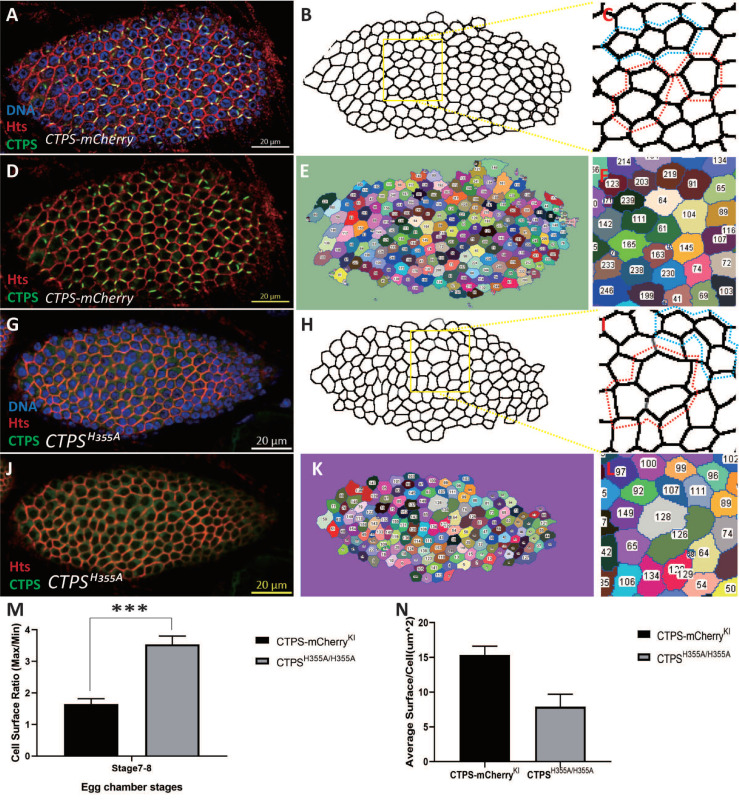
**Quantitative morphological analysis of follicle cells.** (**A**) Surface view of a stage 8 wild-type egg chamber. CTPS-mCherry (green), Hts (red) and DNA (blue, labeled with Hoechst 33342). (**B**) Morphology of cell membrane. The yellow rectangle highlights an area of the follicle epithelium. (**C**) Enlarged part of the framed region in B. Larger follicular cells are circled in red, and smaller follicular cells are circled in blue. (**D**) CTPS-mCherry (green) and Hts (red) of the same egg chamber shown in A. (**E**) The area of each cell was measured after dividing each cell along the cell membrane. (**F**) corresponds to (**D**). (**G**) Surface view of a *CTPS^H355A^* stage 8 egg chamber. CTPS^H355A^-mcherry knock-in (green), Hts (red, labels cell membrane), and DNA (blue, labeled with Hoechst 33342). (**H**) Morphology of the cell membrane. A yellow rectangle highlights an area of the follicle epithelium. (**I**) Enlarged part of the framed region in (**H**). Larger follicle cells are circled in red, and smaller follicle cells are circled in blue. (**J**) CTPS^H355A^-mCherry (green) and Hts (red) of the same egg chamber shown in (E). Surface view of the cell membrane with CTPS. (**K**) The area of each cell was measured after dividing each cell along the cell membrane. (**L**) corresponds to (**K**). Scale bars, 20 μm. (**M**) The ratio of the average area of the three largest cells to the average area of the three smallest cells in an egg chamber. N = 6, ***, *p* < 0.0001. Mann-Whitney U test. (5 stage 7–8 egg chambers/genotypes, 3 biological replicates) (**N**) Average follicle cell surface. N = 3.

**Figure 9 ijms-23-15282-f009:**
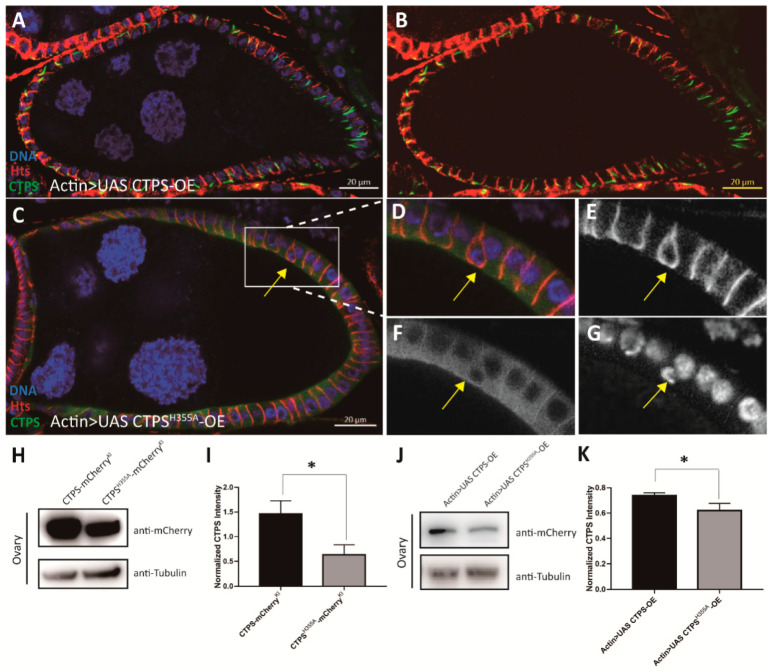
**Follicle cell ingression occurs in egg chamber overexpressing *CTPS^H355A^*.** (**A**,**B**) Side view of stage 8 egg chamber overexpressing *CTPS*. (**C**) Cross section of stage 8 egg chamber overexpressing *CTPS^H355A^*. The ingression is framed by a rectangle. (**D**–**G**) Zoom-in images of the ingression follicle cells in (**C**). Scale bars, 20. (**H**) Western blot is detected with antibodies against mCherry and tubulin on the ovarian lysates of *CTPS-mCh* and *CTPS^H355A^-mCh* mutants. Scale bars, 20 μm. (**I**) Quantitative analysis of the CTPS protein level of samples represented in (**H**), the mean and standard deviation. (**J**) Western blot of the ovarian lysates of *Actin > UAS CTPS-OE* and *Actin > UAS CTPS^H355A^-OE* mutants, detected with antibodies against mCherry and tubulin. (**K**) Quantitative analysis of the CTPS protein level of samples represented in (**J**), the mean and standard deviation. *, *p* < 0.05; Mann-Whitney U test.

**Figure 10 ijms-23-15282-f010:**
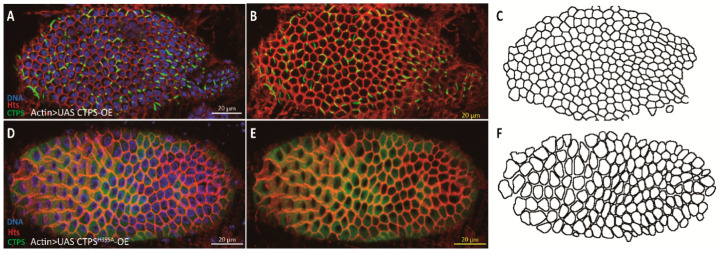
**The integrity of follicle epithelium is compromised when overexpressing *CTPS^H355A^*.** (**A**–**C**) Surface view of a stage 8 egg chamber with follicle cells overexpressing *CTPS*. Large cytoophidia are detectable in almost all follicle cells. (**D**–**F**) Surface view of a stage 8 egg chamber with follicle cells overexpressing *CTPS^H355A^*. Note that the heterogenous sizes of follicle cells and increased gaps between neighbouring follicle cells. CTPS-mCherry (green), Hts (red) and DNA (blue, labelled with Hoechst 33342). Scale bars, 20 μm.

**Figure 11 ijms-23-15282-f011:**
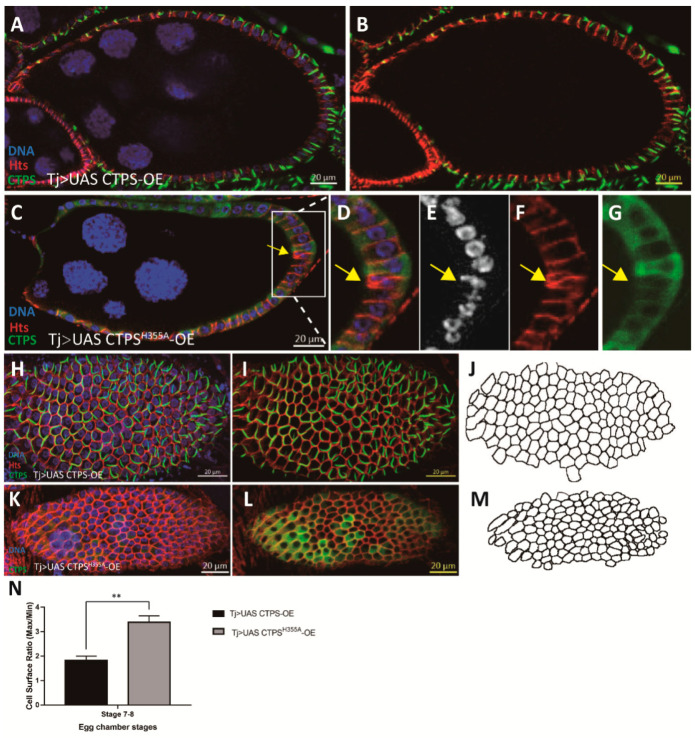
**Follicle cell ingression occurs with follicle cell-specific overexpresson of *CTPS^H355A^*.** (**A**,**B**) Side view of a stage 8 egg chamber overexpressing CTPS in follicle epithelium. (**C**) Cross section of stage 8 egg chamber after overexpression of CTPS^H355A^ in follicle epithelium. The ingression is framed by a rectangle. (**D–G**) Zoom-in of the ingressive follicle cell in (**C**). (**H–J**) Surface view of a stage 8 egg chamber with follicle cell-specific overexpression of CTPS. Large cytoophidia are detectable in almost all follicle cells. Note that most cytoophidia are distributed on or near the cortex of follicle cells. (**K–M**) Surface view of a stage 8 egg chamber with follicle cell-specific overexpression of CTPS^H355A^. No cytoophidium is detectable. CTPS-mCherry (green), Hts (red) and DNA (blue, labelled with Hoechst 33342). Scale bars, 20 μm. (**N**) Quantitative analysis of the ratio of three largest cells versus three smallest cells (5 images/genotypes, 3 biological replicates). Mann-Whitney U test, **, *p* = 0.002.

**Figure 12 ijms-23-15282-f012:**
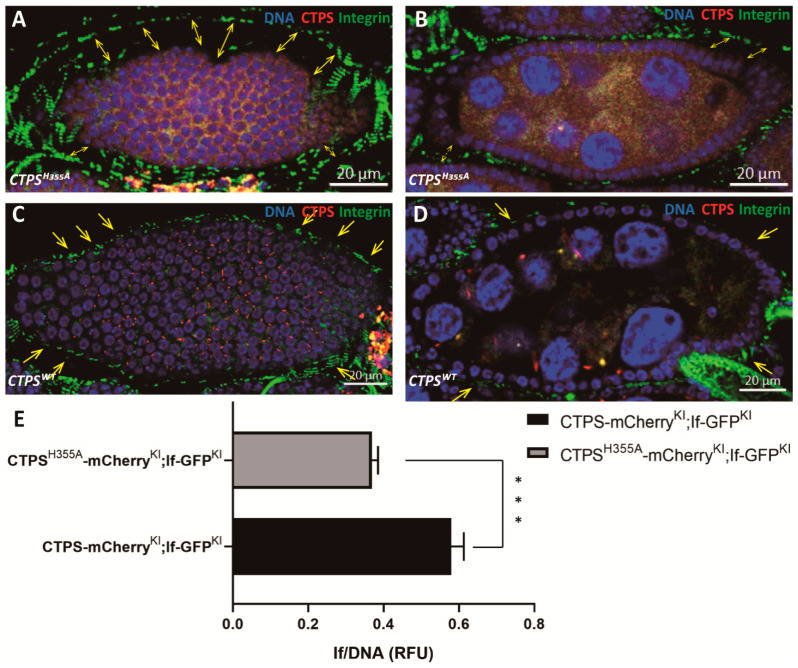
**Compared with wild-type control, the space increases between muscle sheath and egg chambers in *CTPS^H355A^*.** (**A**,**B**) A stage 8 egg chamber of the *CTPS^H355A^* mutant. Integrin is labeled with knock-in GFP (green). CTPS^H355A^ is labeled with knock-in mCherry (red). Yellow arrows point to gap between egg chamber and the muscle sheath. (**C**,**D**) A stage 8 wild-type egg chamber. Integrin is labeled by knock-in GFP. CTPS is labeled by knock-in mCherry. Scale bars, 20 μm. (**E**) The ratio of GFP intensity of the integrin to DNA from (**A**–**D**). The value is normalized to the control (5 images/genotypes, 3 biological replicates). Mann-Whitney U test. ***, *p* = 0.0005.

## Data Availability

Not applicable.

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
