# Peer review of "Cytoophidia Maintain the Integrity of Drosophila Follicle Epithelium"

_ijms, 2022, doi:10.3390/ijms232315282_

Round 1

Reviewer 1 Report

The manuscript of Wang et al. entitled “Cytoophidia maintain the integrity of Drosophila follicle epithelium” deals with the ability of the glutamate aminotransferase CTP synthase (CTPS) to form filamentous structures in different tissues of various organisms. To explore the function of cytoophidia the AA generate transgenic flies with a point mutation in CTPS. This mutation does not affect enzymatic activity but lead to the disassembly of cytoophidia. By IF analysis of Drosophila ovaries with several antibodies, the AA suggest that cytoophidia play important roles in maintaining the integrity of the follicle epithelium. In my opinion, the manuscript is interesting and deserves publication after answering to the minor comments I reported below.

I would like to see control pictures for Figs 2 and 4.

Fig. 4E. The nucleus is unclear.

 Why is the abnormal position of the follicle cell nuclei mainly restricted to the anterior and posterior region of the egg chamber?

 l.163 Not Armidilo but Armadillo

 l.299. Please add reference

 Discussion. It is unclear for me if the Authors report previous data published in Exp. Cell Res. Or if they discuss the present observation. Please make clear this point.

 Please, carefully check references. For example, in l.431 I see the complete page numbers (1497-1503) but in l.434 (113-6).

Author Response

Reviewer #1 (Round 1):
The manuscript of Wang et al. entitled “Cytoophidia maintain the integrity of
Drosophila follicle epithelium” deals with the ability of the glutamate
aminotransferase CTP synthase (CTPS) to form filamentous structures in
different tissues of various organisms. To explore the function of cytoophidia
the AA generate transgenic flies with a point mutation in CTPS. This mutation
does not affect enzymatic activity but lead to the disassembly of cytoophidia.
By IF analysis of Drosophila ovaries with several antibodies, the AA suggest
that cytoophidia play important roles in maintaining the integrity of the follicle
epithelium. In my opinion, the manuscript is interesting and deserves
publication after answering to the minor comments I reported below.
We would conduct more extensive experiments to
I would like to see control pictures for Figs 2 and 4.
We have updated control figures for Figs 2 and 4.
Fig. 4E. The nucleus is unclear.
Figure 4E have updated.
Why is the abnormal position of the follicle cell nuclei mainly restricted to the
anterior and posterior region of the egg chamber?
It looks like follicle cells near the posterior region are more likely to ingress, but
we are still working on this question. One propose is that an egg chamber is an
ellipsoid structure with the highest curvature of the ellipsoid at both ends, so the
follicle cells located at these positions may be subjected to a greater squeezing
force. Due to the absence of cytoophidia that help maintain cellular structure,
these follicle cells are likely to be squeezed inwards.
l.163 Not Armidilo but Armadillo
Corrected.
l.299. Please add reference
Added.
Discussion. It is unclear for me if the Authors report previous data published in
Exp. Cell Res. Or if they discuss the present observation. Please make clear
this point.
2 / 2
Previous data has been published in ECR, and we have now added the
reference.
Please, carefully check references. For example, in l.431 I see the complete
page numbers (1497-1503) but in l.434 (113-6).
We have changed all the references to show the full page numbers.

Reviewer 2 Report

Here the authors investigate the role of cytoophidia in Drosophila egg chambers.  The enzyme CTPS has a function in the production of CTP, but also forms novel filamentous structures called cytoophidia. Throughout the paper, they show high quality imaging and analysis.

Using a knockin of the H355A mutation, they observe a loss of cytoophidia and diffuse localization of CTPS.  They find defects in the egg chambers ranging from ingressing follicle cells to cell shape changes.  While the cell shape changes are nicely quantified, quantification is notably missing from the ingression phenotype and should be provided. Overall they do an excellent job of finding subtle phenotypes, but a further description of the impact on oogenesis such as border cell migration, cell death of egg chambers or impacts on fecundity would strengthen the paper. 

Concerns that should be addressed:

1.     It is important to confirm that the H355A mutation or the mCherry tag does not affect the CTP synthase function – this is stated but there are no data provided or a reference to this point. 

2.     The ingression phenotype is interesting, but the effect is subtle and no quantification is provided. It is not clear how often this phenotype is observed. Quantification should be provided with the mCherry wt knockin as a control.

Additional comments to consider:

3.     Interestingly, most of the images they show are posterior follicle cells – are these the polar cells?  Is polar cell number or border cell migration disrupted?  The BCs look abnormal in Figure 5.

4.     It would be a good idea to check on Wolbachia status of the mutant and control given that Wolbachia infects follicle cells and could be a confounding factor.

5.     What is meant by the genotype CTPSH355A/- In figure 6? 

Author Response

Authors’ response (in blue color)

Reviewer #2 (Round 1):

This is an interesting study in which the investigators explore the effect of cytoophidia formation on follicle cell function. A number of striking images of stained egg chambers are presented. Limitations of the study include unclear description of strains, a lack of controls for genetic background, lack of controls for effects of the mCherry protein tag, and lack of details on quantification and statistical analyses.

  1. The H355 point mutation is stated to disrupt cytoophidia formation. However, the investigators also modified the protein by adding an mCherry moiety at the C-terminus. It is not clear how do the authors know that the altered function of the mutant protein is due to the point mutation and the failure to form cytoophidia, and not due to some interfering effect of the mCherry moiety. I think to address this would require analysis of an mCherry-tagged wild-type protein, to confirm that it has a normal phenotype, and that it can rescue a CTPS null mutation.

Our lab generated the CTPS-mCherryKI Drosophila strain out of the w1118 Drosophila strain. We have added an additional Figure (new Figure 2) to show that CTPS could form cytoophidia in CTPS-mCherryKI Drosophila strain, which have the same distribution pattern as those in w1118. Also, we have added two previous studies in our lab as references, which used CTPS-mCherryKI Drosophila strain as the wild-type control (Refs 21, 22).

  1. The Figure 6 morphological comparison of follicles – the description of the experiment and quantification is not clear.

First - it is not clear that the w[1118] strain and the mutant strain have the same genetic background, as no backcrossing or other strategies to control genetic background are mentioned in the Methods sections. Because of this, we cannot be certain that the difference observed between the w[1118] strain and the mutant is actually due to the mutation, as opposed to being due to other differences in genetic background.

As mentioned before, our lab generated the CTPS-mCherryKI Drosophila strain out of the w1118 Drosophila strain. Afterwards, our lab generated the CTPSH355A-mCherryKI Drosophila strain out of the CTPS-mCherryKI Drosophila strain.

Second – the quantification and statistical analysis is not clear. Line 169 states “there was a significant difference (p<0.05) in the number of heptagonal follicle cells in different stages of egg chambers of the CTPSH355A mutant.” But I can find no analysis of different stage egg chambers, and no information on how the p value was generated.

We have focused our analysis to stage 8 egg chambers and revised the statement accordingly.

Line 171 states “Moreover, the number of hexagonal follicle cells in CTPSH355A egg chambers decreased by about 15% at stage 8, while the number of polygonal cells increased by about 10% (Figure 6E-G).” However, Fig 6G does not indicate how % values were obtained, how any replication was employed, or how p values were generated.

We used confocal microscopy to obtain the images of the surface of egg chambers and then counted different shapes cell by cell. Six stage 8 egg chambers per time, together with three biological repeats. We could use histogram to show the significance. However, in order to show the effects more intuitively, we changed the histogram to a line chart. The generation of p values is now described in the figure legends.

  1. Figure 7 reports analysis of a “CTPS-mCherry” expressing line used as a control for analysis of cell size. This appears to be the control that is missing from the experiments on cell introgression, mentioned above in comment #1. I think the authors need to show that this construct can rescue a CTPS null mutant, to demonstrate that the mCherry moiety on the wild-type and the H355 point mutant is not causing an unintended phenotype(s).

We have added an additional Figure (new Figure 2) to show that CTPS could form cytoophidia in CTPS-mCherryKI Drosophila strain, which have the same distribution pattern as those in w1118. Also, we have added two previous studies in our lab as references, which used CTPS-mCherryKI Drosophila strain as the wild-type control (Refs 21, 22).

  1. Several fly strains are not sufficiently described. For example, multiple strains, such as tj-GAL4, are used without describing the source of the strains, or giving references to support the patterns of expression described for the strains.

Because we used UAS CTPS-mCherry Drosophila strains with fluorescent protein mCherry to hybridize with Tj-Gal4 as well as Actin-gal4, the expression pattern of fluorescent protein mCherry is the expression pattern of gal4 driver respectively. We observed the expression of fluorescent proteins by confocal microscopy and confirmed that the expression patterns of the two Gal4 drivers are correct.

Reviewer 3 Report

This is an interesting study in which the investigators explore the effect of cytoophidia formation on follicle cell function. A number of striking images of stained egg chambers are presented. Limitations of the study include unclear description of strains, a lack of controls for genetic background, lack of controls for effects of the mCherry protein tag, and lack of details on quantification and statistical analyses.

1. The H355 point mutation is stated to disrupt cytoophidia formation. However, the investigators also modified the protein by adding an mCherry moiety at the C-terminus. It is not clear how do the authors know that the altered function of the mutant protein is due to the point mutation and the failure to form cytoophidia, and not due to some interfering effect of the mCherry moiety. I think to address this would require analysis of an mCherry-tagged wild-type protein, to confirm that it has a normal phenotype, and that it can rescue a CTPS null mutation.

2. The Figure 6 morphological comparison of follicles – the description of the experiment and quantification is not clear.

First -  it is not clear that the w[1118] strain and the mutant strain have the same genetic background, as no backcrossing or other strategies to control genetic background are mentioned in the Methods sections. Because of this, we cannot be certain that the difference observed between the w[1118] strain and the mutant is actually due to the mutation, as opposed to being due to other differences in genetic background.

Second – the quantification and statistical analysis is not clear. Line 169 states “there was a significant difference (p<0.05) in the number of heptagonal follicle cells in different stages of egg chambers of the CTPSH355A mutant.” But I can find no analysis of different stage egg chambers, and no information on how the p value was generated. Line 171 states “Moreover, the number of hexagonal follicle cells in CTPSH355A egg chambers decreased by about 15% at stage 8, while the number of polygonal cells increased by about 10% (Figure 6E-G).” However, Fig 6G does not indicate how % values were obtained, how any replication was employed, or how p values were generated.

3. Figure 7 reports analysis of a “CTPS-mCherry” expressing line used as a control for analysis of cell size. This appears to be the control that is missing from the experiments on cell introgression, mentioned above in comment #1. I think the authors need to show that this construct can rescue a CTPS null mutant, to demonstrate that the mCherry moiety on the wild-type and the H355 point mutant is not causing an unintended phenotype(s).

4. Several fly strains are not sufficiently described. For example, multiple strains, such as tj-GAL4, are used without describing the source of the strains, or giving references to support the patterns of expression described for the strains.

Author Response

Authors’ response (in blue color)

Reviewer #3 (Round 1):

Here the authors investigate the role of cytoophidia in Drosophila egg chambers.  The enzyme CTPS has a function in the production of CTP, but also forms novel filamentous structures called cytoophidia. Throughout the paper, they show high quality imaging and analysis.

Using a knockin of the H355A mutation, they observe a loss of cytoophidia and diffuse localization of CTPS.  They find defects in the egg chambers ranging from ingressing follicle cells to cell shape changes.  While the cell shape changes are nicely quantified, quantification is notably missing from the ingression phenotype and should be provided.

Overall they do an excellent job of finding subtle phenotypes, but a further description of the impact on oogenesis such as border cell migration, cell death of egg chambers or impacts on fecundity would strengthen the paper.

Many thanks for the kind words and suggestions. We will investigate the effects of CTPSH355A on oogenesis in our follow-up studies.

Concerns that should be addressed:

  1. It is important to confirm that the H355A mutation or the mCherry tag does not affect the CTP synthase function – this is stated but there are no data provided or a reference to this point. 

We have added a new figure (new Figure 2) to show that CTPS can form cytoophidia in CTPS-mCherryKI Drosophila strain, with identical pattern as that in the wild-type. Additional references on H355A have been added.  

  1. The ingression phenotype is interesting, but the effect is subtle and no quantification is provided. It is not clear how often this phenotype is observed. Quantification should be provided with the mCherry wt knockin as a control.

We have added the ingression frequency in Fig 4.

Additional comments to consider:

  1. Interestingly, most of the images they show are posterior follicle cells – are these the polar cells? Is polar cell number or border cell migration disrupted? The BCs look abnormal in Figure 5.

We thought polar cells are possibly to ingress but not all ingressive cells are polar cells. We haven’t found the abnormality in border cell migration.

  1. It would be a good idea to check on Wolbachia status of the mutant and control given that Wolbachia infects follicle cells and could be a confounding factor.

Thanks for the suggestion and we love to look into this in the future.

  1. What is meant by the genotype CTPSH355A/- In figure 6?

We have corrected it. It’s the heterozygous CTPSH355A/TM6B strain.

Round 2

Reviewer 2 Report

The authors partially addressed my previous concerns.  They showed that the mCherry tag does not affect function and they provided some quantification of the phenotype.

However, they still did not address whether the H355A mutation affects the CTP synthase function or if it is specific for cytoophidia formation.  Without a statement addressing this, it is possible that the observed phenotype is due to the CTP synthase function and this needs to be acknowledged. 

In addition, while they did provide quantification, no statistics were performed.  At minimum, the number of egg chambers examined for the quantification should be stated.

Author Response

Authors’ response (in blue color)

Reviewer #2 (Round 2):

The authors partially addressed my previous concerns. They showed that the mCherry tag does not affect function and they provided some quantification of the phenotype.

However, they still did not address whether the H355A mutation affects the CTP synthase function or if it is specific for cytoophidia formation.  Without a statement addressing this, it is possible that the observed phenotype is due to the CTP synthase function and this needs to be acknowledged.

The amino acid histidine at the 355th position, or His355, lies at the tetramer-tetramer interface of CTPS protein (See Ref 22). If the H355 site is mutated, the cytoophidium cannot be formed. Previous studies showed that the H355 site is essential for its polymerization, but not enzymatic function (See Ref 23, 24). Our laboratory has solved the structure of dmCTPS and found that the H355 site lies at the tetramer-tetramer interface that doesn’t affect the catalytic site (See Ref 25).

In addition, while they did provide quantification, no statistics were performed.  At minimum, the number of egg chambers examined for the quantification should be stated.

As we have mentioned in the figure legends as well as methods, for the frequency of ingression, we randomly collected images of 20 egg chambers per genotype. For the morphology changes of follicle cells, we collected the surface images using Zeiss 880, z-stack interval as 0.5um of more than 5 stage 8 egg chambers one time per genotype to count cells of different shapes and sizes, biological repeats = 3. Mann-Whitney U test was conducted to compare two groups of data.

Reviewer 3 Report

The author did not address the comments from the first review.

1. They still have not confirmed that the mCherry moeity does not disrupt the normal function of the CTPS.

2. Genetic background is not controlled and is not well described.

3. There is still no clear description of heptagonal cell quantification and how p values were generated. 

4. The authors still do not provide the source of the Tj-Gal4 and Actin-Gal4 lines, or any references, or any controls for expression patterns.

Author Response

Authors’ response (in blue color)

Reviewer #3 (Round 2):

The author did not address the comments from the first review.

  1. They still have not confirmed that the mCherry moeity does not disrupt the normal function of the CTPS.

In order to find out whether the abnormality is caused by the inability of CTPS to aggregate due to H355A point mutation or the addition of mCherry tag, our laboratory constructed another Drosophila strain with mCherry added to the C-terminus of CTPS based on w1118. To determine whether the feature of cytoophidium localization was artifact introduced by protein fusion between CTPS and mCherry tag, we performed immunofluorescence microscopy and directly detected CTPS protein of the w1118 fly and found no difference (See Ref 26). It is proved by the observation that the knock-in mCherry tag does not affect the polymerization of CTPS protein. The morphology of the CTPS-mCherryKI Drosophila ovaries is consistent with that of the w1118, which implies that the CTPS-mCherryKI Drosophila can also be used as control in our experiment (Figure 2 A-F). Also, our laboratory has used the CTPS-mCherryKI Drosophila as control in previous studies (See Ref 27, 28).

  1. Genetic background is not controlled and is not well described.

Transgenic flies expressing full-length CTPS (isoform C, LD25005) under the UAS 386 promoter (UAS-CTPS) were generated in our laboratory (Back cross 5 generations before use).

  1. There is still no clear description of heptagonal cell quantification and how p values were generated.

All images were obtained under laser-scanning confocal microscopy (Zeiss 880). We used the ImageJ SCF function to segment the follicle cells by the cell membrane, then use ImageJ cell counter to get the number of polygonal cells. We also use ImageJ to measure the area of each cell. For each statistical quantification, we collected the surface images using Zeiss 880, z-stack interval as 0.5um of more than 5 stage 8 egg chambers one time per genotype, biological repeats = 3. Mann-Whitney U test was conducted to get the p-value.

  1. The authors still do not provide the source of the Tj-Gal4 and Actin-Gal4 lines, or any references, or any controls for expression patterns.

We added references in the materials and methods part. As Actin-Gal4/Cyo (A gift from Guanjun Gao’s 378 laboratory, ubiquitous expression under strong promoter, a chromosome II insertion balanced over Curly of Oster (See Ref 30) ) and Tj-Gal4 (A gift from Kun Dou’s laboratory (See Ref 31,32)).

Round 3

Reviewer 2 Report

The authors have addressed my concerns.

Author Response

We would like to take this opportunity to thank you for your comments and your time.

Reviewer 3 Report

The authors have still not addressed the main comments from the first round of reviews.

Limitations of the study include a lack of controls for genetic background, lack of controls for effects of the mCherry protein tag, and lack of details on quantification and statistical analyses.

1. The H355 point mutation is stated to disrupt cytoophidia formation. However, the investigators also modified the protein by adding an mCherry moiety at the C-terminus. It is not clear how do the authors know that the altered function of the mutant protein is due to the point mutation and the failure to form cytoophidia, and not due to some interfering effect of the mCherry moiety. I think to address this would require analysis of an mCherry-tagged wild-type protein, to confirm that it has a normal phenotype, and that it can rescue a CTPS null mutation.

2. The Figure 7 morphological comparison of follicles – the description of the experiment and quantification is not clear.

The w[1118] strain and the mutant strain do not have the same genetic background, as the construction descried in the Methods sections does not control genetic background between strains. Because of this, we cannot be certain that the difference observed between the w[1118] strain and the mutant is actually due to the mutation, as opposed to being due to other differences in genetic background.

Second – the quantification and statistical analysis is not clear. Line 205 states “Moreover, the number of hexagonal follicle cells in CTPSH355A egg chambers decreased by about 15% at stage 8, while the number of polygonal cells increased by about 10% (Figure 7E-G).” However, Fig 7G does not indicate how % values were obtained, how any replication was employed, or how p values were generated.

3. Figure 8 reports analysis of a “CTPS-mCherry” expressing line used as a control for analysis of cell size. This appears to be the control that is missing from the experiments on cell introgression, mentioned above in comment #1. I think the authors need to show that this construct can rescue a CTPS null mutant, to demonstrate that the mCherry moiety on the wild-type construct, and on the H355 point mutant construct, is not causing an unintended phenotype(s).

Author Response

Authors’ response 

Reviewer #3 (Round 3):

The authors have still not addressed the main comments from the first round of reviews.

Limitations of the study include a lack of controls for genetic background, lack of controls for effects of the mCherry protein tag, and lack of details on quantification and statistical analyses.

  1. The H355 point mutation is stated to disrupt cytoophidia formation. However, the investigators also modified the protein by adding an mCherry moiety at the C-terminus.

As mentioned before, in order to find out whether the abnormality is caused by the inability of CTPS to aggregate due to H355A point mutation or the addition of mCherry tag, our laboratory constructed another Drosophila strain with mCherry added to the C-terminus of CTPS based on w1118.

It is not clear how do the authors know that the altered function of the mutant protein is due to the point mutation and the failure to form cytoophidia, and not due to some interfering effect of the mCherry moiety.

To determine whether the feature of cytoophidium localization was artifact introduced by protein fusion between CTPS and mCherry tag, we performed immunofluorescence microscopy and directly detected CTPS protein of the w1118 fly and found no difference of the cytoophidia morphology as well as distribution (See Ref 26). Hence our laboratory has confirmed that the w1118 fly and the CTPS-mCh fly are interchangeable for morphological analysis, both of which have the same features as control. Also, our laboratory has used the CTPS-mCherryKI Drosophila as control in previous studies (See Refs 27, 28).

Furthermore, the structure of dmCTPS solved by our laboratory showed that residues556-627 are disordered in the dmCTPS monomers, which are already not the possible effectors for CTPS monomers polymerization (See Ref 25). We added an mCherry tag to the C-terminus of CTPS and leave the origin residues556-627 unchanged should also not affect the polymerization.

I think to address this would require analysis of an mCherry-tagged wild-type protein, to confirm that it has a normal phenotype,

It is proved by the observation that the knock-in mCherry tag does not affect the polymerization of CTPS protein. The morphology of the CTPS-mCherryKI Drosophila ovaries is consistent with that of the w1118 (See Ref 26,27,28), which implies that the CTPS-mCherryKI Drosophila can also be used as control in our experiment (Figure 2 A-F).

and that it can rescue a CTPS null mutation.

The CTPS null mutation is lethal so we couldn’t perform that experiment.

  1. The Figure 7 morphological comparison of follicles – the description of the experiment and quantification is not clear.

The w[1118] strain and the mutant strain do not have the same genetic background, as the construction descried in the Methods sections does not control genetic background between strains. Because of this, we cannot be certain that the difference observed between the w[1118] strain and the mutant is actually due to the mutation, as opposed to being due to other differences in genetic background.

We have addressed that our laboratory has confirmed that the W1118 fly and the CTPS-mCh fly are interchangeable for morphological analysis, both of which have the same features as control (See Ref 25). Also, our laboratory has used the CTPS-mCherryKI Drosophila as control in previous studies (See Refs 27, 28). 

For the concern of genetic background, our lab generated the CTPS-mCherryKI Drosophila strain out of the w1118 Drosophila strain. We have added an additional Figure (new Figure 2) to show that CTPS could form cytoophidia in CTPS-mCherryKI Drosophila strain, which have the same distribution pattern as those in w1118.

Second – the quantification and statistical analysis is not clear. Line 205 states “Moreover, the number of hexagonal follicle cells in CTPSH355A egg chambers decreased by about 15% at stage 8, while the number of polygonal cells increased by about 10% (Figure 7E-G).” However, Fig 7G does not indicate how % values were obtained, how any replication was employed, or how p values were generated.

All images were obtained under laser-scanning confocal microscopy (Zeiss 880). Because follicle cells are like polygonal prisms covering the ellipsoid egg chamber, we could captured the base of follicle cells using Z-stack function of the Microscopy. We used the ImageJ SCF function to segment the follicle cells by the cell membrane, then use the ImageJ cell counter to get the number of polygonal cells. After getting the number of cells in different shapes, we use Excel to add them up to get the total amount of cells on the surface of one egg chamber we captured, then we do the math to get the percentage of different polygonal cells. For each statistical quantification, we collected the surface images using Zeiss 880, z-stack interval as 0.5um of more than 5 stage 8 egg chambers one time per genotype, biological repeats=3. Mann-Whitney U test was conducted using GraphPad to get the p-value.

  1. Figure 8 reports analysis of a “CTPS-mCherry” expressing line used as a control for analysis of cell size. This appears to be the control that is missing from the experiments on cell introgression, mentioned above in comment #1. I think the authors need to show that this construct can rescue a CTPS null mutant, to demonstrate that the mCherry moiety on the wild-type construct, and on the H355 point mutant construct, is not causing an unintended phenotype(s).

For the concern of the mCherry moiety, we first confirmed that the CTPS knock-in mCherry in w1118 Drosophila had no effect on the aggregation of CTPS and the morphology as well as distribution of the cytoophidia (See Refs 26-28).

In the following experiments, the overexpression of wild-type CTPS with an mCherry tag was driven by the Actin-Gal4 driver, and the difference between the overexpression of CTPS-mCh and CTPSH355A-mCh was obtained, which should further rule out the effects caused by the expression of mCherry.
